# Active Surveillance in Non-Muscle Invasive Bladder Cancer: A Systematic Review

**DOI:** 10.3390/cancers17101714

**Published:** 2025-05-20

**Authors:** Míriam Campistol, Fernando Lozano, Albert Carrion, Carles Xavier Raventós, Juan Morote, Enrique Trilla

**Affiliations:** 1Department of Urology, Vall d’Hebron Hospital, Passeig de la Vall d’Hebron 119, 08035 Barcelona, Spain; fernando.lozano@vallhebron.cat (F.L.); albert.carrion@vallhebron.cat (A.C.); carlesxavier.raventos@vallhebron.cat (C.X.R.); juan.morote@vallhebron.cat (J.M.); enrique.trilla@vallhebron.cat (E.T.); 2Department of Surgery, Universitat Autonoma de Barcelona, 08193 Barcelona, Spain; 3Research Unit in Biomedicine and Translational Oncology, Research Institute Vall Hebron University Hopital (VHIR), 08035 Barcelona, Spain

**Keywords:** non-muscle invasive bladder cancer, active surveillance, expectant management, TURBT, cystoscopy

## Abstract

Bladder cancer is a common disease, especially in older adults. Most cases are found early, when the cancer is limited to the inner layer of the bladder. The standard treatment involves transurethral resection of the tumor. While this method is effective, it can lead to repeated procedures, discomfort, and high medical costs. As a result, doctors and researchers are exploring a different approach called active surveillance. This means carefully monitoring patients with low-risk tumors through regular check-ups, rather than operating right away. In this study, we reviewed the results of several previous studies to understand how safe and effective active surveillance is for patients with early-stage bladder cancer. We found that for carefully chosen patients, active surveillance can be a safe and practical option. Most people under observation did not experience serious tumor growth or spread, and many were able to avoid surgery. However, there is still no agreement on the best way to decide who should be monitored or how often check-ups should occur. More research is needed to create clear rules. Using active surveillance could reduce medical risks and improve quality of life for some people with bladder cancer.

## 1. Introduction

Bladder cancer is the ninth most common diagnosed cancer worldwide and ranks as the thirteenth leading cause of cancer-related mortality [1]. Approximately 70%–80% of newly diagnosed cases of bladder cancer are classified as non-muscle invasive (NMIBC), comprising Ta (70%), T1 (20%) and isolated carcinoma in situ (CIS) (10%) [2,3]. Although the majority of patients with NMIBC experience recurrence during follow-up, only 15% will progress to muscle-invasive or metastatic disease, with a mortality rate at five years below 1% [4]. Nonetheless, clinical evidence indicates that patients with small (≤5 mm), low-grade tumors have an even lower risk of progression and mortality [5].

Recent advances in the molecular understanding of bladder cancer suggest that low-grade, non-muscle-invasive tumors represent a biologically distinct entity with a separate pathogenesis from high-grade tumors. The biological characteristics of the initial tumor remain the most significant predictor of long-term outcomes [6]. The low-grade pathway is thought to originate from benign urothelium through a process of urothelial hyperplasia. In contrast, the majority of invasive tumors are likely to originate from the progression of dysplasia to carcinoma in situ and papillary high-grade noninvasive or invasive cancer [7]. This distinction could explain why low-grade tumor recurrences usually have the same histological features and the difference in risk of progression between both groups [8]. Indeed, Catto et al. demonstrated that miRNA expression profiles share more alterations between high-grade NMIBC and MIBC than between low-grade and high-grade NMIBC tumors, suggesting that despite the absence of muscle-invasion at the time of TURBT, high-grade NMIBC is a clear precursor to muscle-invasive disease, and thus warrants a more aggressive and intensive therapeutic approach akin to that used for MIBC [9].

The current European Association of Urology guidelines [10] recommend stratifying patients into four categories, providing tailored treatment recommendations based on the risk of recurrence and/or progression. Low-risk patients are defined as those with small (<3 cm), solitary, low grade Ta tumors without CIS and should receive an immediate installation of chemotherapy following transurethral resection of the bladder tumor (TURBT). High and very high-risk patients include those with T1, G3, CIS or multiple, recurrent and large (>3 cm) TaG1-G2 tumors. These patients should undergo full-dose Bacillus Calmette-Guérin (BCG) installations for 1–3 years [11,12] or be considered for radical cystectomy in very high-risk cases. Intermediate-risk patients, comprising all cases not falling within the low-, high-risk or very high-risk categories, should receive adjuvant therapy with BCG (1 year or more) or further installation of chemotherapy (6 to 12 months) [13].

Cystoscopy remains the gold-standard technique for the diagnosis of bladder cancer, and its findings have very good correlation with grade and stage in the hands of experienced urologists [14]. If detected, a TURBT is performed to remove the lesion and assess its stage and grade. TURBT serves as both, the definitive diagnostic procedure for bladder cancer and also represents the most important therapeutic tool for patients with NMBIC [15]. While TURBT has been the preferred technique for decades and is currently the gold-standard treatment for NMIBC nowadays. Despite its low morbidity rates, it has several notable limitations. The risk of under-staging T1 tumors ranges from 8% to 11% depending on the study, while the risk of persistence may reach up to 51% to 58% [16,17]. Approximately 4–6% of patients develop urinary tract infections or transient hematuria after the procedure. Sever complications, though less frequent, include transurethral resection syndrome, active hematuria requiring endoscopic hemostasis or blood transfusion, bladder wall perforation and urethral strictures [18,19]. Oncologic outcomes can also be compromised due to factors such as thermal damage at the lesion margins, absence of detrusor muscle, tumor cell seeding and reimplantation or incomplete resection [18]. Furthermore, the high recurrence rates associated with bladder cancer, combined with the complications of TURBT and the need for repeated cystoscopies during long-term follow-up, impose a significant burden on patients. These factors make bladder cancer one of the costliest malignancies in terms of both economic impact and quality of life [20]. To mitigate these challenges, reducing the frequency of follow-up procedures such as cystoscopy and TURBT in patients with low-grade, low-risk NMIBC is essential. This approach could decrease the patient burden, improve compliance and minimize procedure-related complications. Tailored strategies based on individual patient characteristics and tumor biology could help achieve an optimal balance between effective disease monitoring and minimizing both patient burden and healthcare costs. 

Active surveillance is defined as a treatment approach that involves closely monitoring a patient’s condition but not giving a treatment unless test results indicate the condition is worsening. The aim of AS is to avoid overtreatment and delay the need for intervention, while still allowing the opportunity for curative treatment if necessary, during ongoing, close follow-up. 

Urologists are well-acquainted with AS for urogenital malignancies, including low-risk prostate cancer [21] and small renal masses [22]. Although AS for bladder cancer was initially reported more than two decades ago [23], it remains an uncommon strategy. 

In 2003, Soloway et al. [23] suggested AS for selected recurrent low-grade NMIBC as a safe alternative to TURBT and, despite the concept of AS still being relatively new in the NMIBC field, it was subsequently widely adopted and included in international guidelines as a treatment option for low-grade NMIBC [24], making the strategy useful for low-grade bladder cancer and some cases of intermediate-risk disease. Nevertheless, there is still a need for new protocols in order to better identify the most suitable patients for AS and predict both the risk of AS failure and disease progression [5]. 

There is currently no consensus on the optimal approach AS or the ideal frequency of cystoscopies. However, most studies recommend intensive follow-up during the first year of AS, with cystoscopies performed every 3 months, as this period represents the highest rate of discontinuation from AS. In some cases, this frequency is extended to a second year and then reduced to biannual cystoscopies [25,26]. One of the primary advantages of AS is its potential to reduce the need for surgical interventions. When a low-grade NMIBC recurrent tumor is detected and excised, there is a significant likelihood of subsequent recurrences, which leads to more surgeries [27,28]. By adopting AS at the time of initial recurrence, subsequent surgical interventions may be avoided, thereby reducing the overall surgical burden on patients. 

Considering the limited evidence of the optimal criteria for selecting patients for AS, we undertook a comprehensive systematic review and a pool analysis of the current available evidence, to evaluate oncological outcomes, defined as failure rate, upstaging and grade progression. 

## 2. Acquisition of Evidence

### 2.1. Search Strategy and Results

A comprehensive literature search was conducted in the PubMed, Cochrane and Trip databases by two independent reviewers in December 2024 using the Medical Subject Heading (MeSH) terms: “active surveillance” and/or “expectant management”, “bladder tumor” and/or “bladder cancer” [29]. While only studies written in English were included in the analysis, the search was not restricted by language. To refine the results, the following Boolean operators were applied: (“bladder cancer” OR “bladder tumor”) AND (“active surveillance” OR “expectant management”). The Preferred Reporting Items for Systematic Reviews and Meta-Analyses (PRISMA) guidelines were followed [30], and the Population, Intervention, Comparison and Outcomes (PICO) selection criteria [31] were employed. The selected population comprised patients with low-grade NMIBC who underwent AS, compared with those treated with TURBT. The primary outcome was tumor progression. This systematic review was registered in the PROSPERO database (ID 1035723).

Original articles and studies with prospective and retrospective design were included in this review. Thus, letters, editorial, case reports, protocols, narrative reviews, poster or oral communications, and prevalence studies were excluded. A total of 69 articles containing the MeSH terms previously defined were identified in PubMed. Based on title screening, 33 were excluded. Specifically, 10 analyzed the epidemiology and survival of other types of cancers, 5 investigated different aspects of prostate pathology, 8 focused on MIBC, 4 addressed multiple imaging tools, 2 were case reports and 3 discussed kidney cancer. 

Based on the evaluation of abstracts, 36 articles were selected for full-text review according to the topic of interest. Finally, following the PRISMA guidelines and PICO criteria, 11 original articles were selected for evidence synthesis. The PRISMA flowchart summarizing the selection process is summarized in Figure 1. 

### 2.2. Quality Assessment

The Newcastle–Ottawa quality assessment scale was used to analyze the quality of the included studies, and evaluate the risk of bias in the study population selection, whether confounding factors were controlled (comparability) and how well outcomes were measured and followed (outcome) [32]. 

Most studies exhibited a moderate risk of bias, with well-defined selection criteria, reliable outcome assessment methods, and adequate follow-up duration and completeness. However, the absence of a comparison group in the majority of the studies, along with unclear adjustment for confounding factors, limits their robustness. The detailed quality assessment results are presented in Table 1. 

### 2.3. Data Extraction 

The selected studies were thoroughly examined and the following information was extracted from each article: name of first author, year of publication, type of study, patients included, inclusion and exclusion criteria, previous intravesical therapy (BCG or Mitomycin C), median follow-up, median time of AS, grade and stage progression and number of patients who progressed to MIBC.

## 3. Evidence Synthesis

A total of 11 articles that met the PICO criteria previously established were included in this systematic review. Most relevant characteristics are presented in Table 2. 

Soloway et al. [23] (2003) published the first study to investigate observation as an alternative to TURBT for patients with recurrent NMIBC. This retrospective study included 32 patients with small (<1 cm), papillary, low-grade tumors and a history of Ta or T1 bladder cancer. The median follow-up period was 38 months, with cystoscopies conducted every 3 to 6 months. TURBT was performed based on tumor growth, chances in tumor appearance, or gross hematuria. Notably, only 6.7% of cases progressed to high-grade lesions, and no patients developed a MIBC. Thus, the authors concluded that immediate intervention of small, recurrent, low-grade appearing bladder tumors may not always be required.

Pruthi et al. [33] conducted a retrospective analysis a cohort consisting of 22 patients with 32 AS periods with a mean follow-up time of 25 months. The study included patients diagnosed with non-muscle-invasive low-grade urothelial carcinoma, stage Ta, T1 or CIS. All participants had a history of recurrent low-risk bladder cancer and were monitored through cystoscopies every 3 months for the first 2 years, then every 6 months up to 5 years, and annually thereafter. Of the 22 patients, 15 (68.8%) require no intervention, 3 (14%) underwent office fulguration and 4 (18%) required repeat a TURBT. Additionally, 2 patients (9%) showed evidence of stage progression and 1 (4.5%) exhibited grade progression. The authors concluded that AS is a viable option for certain patients with a history of Ta, low-grade tumors, as it helps void the potential risks and comorbidities associated with TURBT. 

Gofrit et al. [42] conducted a retrospective study with 31 patients undergoing 43 AS periods, with a mean follow-up of 18 months. Cystoscopy and bladder-washing cytology were performed every 3 months for 2 years and then every 6 months. The inclusion criteria of the study were as follows: prior resection of low-grade (G1 or G2) pTa tumors, no history of high-grade (G3) tumors, asymptomatic, small (<10 mm) papillary tumors and negative urinary cytology. Among these periods, 35 (81.4%) concluded with tumor resection, all classified as low-grade pTa except a single case of stage T1 and one CIS. The primary reasons for discontinuing surveillance were the development of additional tumors (34.9%) and a significant tumor growth (27.9%). They concluded that following up without TURBT is feasible in previously resected small, recurrent papillary low-grade tumors. 

Hernández et al. [35] published a retrospective analysis in 2009 evaluating patients with NMIBC who underwent TURBT. The study included 64 patients, accounting for a total of 70 observation periods. The inclusion criteria were recurrent small (<10 mm), papillary tumors with a prior diagnosis of NMIBC (low-grade pTa or pT1a), and fewer than five tumors. Patients with a history of high-grade disease, CIS or positive urinary cytology were excluded. The mean follow-up period was 38.6 months with an average observation duration of 10.3 months. Cystoscopy and washing urine cytology were performed every 3–4 months. The most frequent reasons for terminating observation were an increase in size and/or number of tumors. Notably, only three patients progressed to high-grade tumor or CIS and none progressed to MIBC. These findings suggested that AS can be safely offered to patients with specific characteristics, with minimal risk of stage or grade progression, thereby reducing the need of surgical intervention. 

The same authors published an updated analysis of the same patient cohort, extending the study in time and including a total of 186 individuals with a prior history of low-grade NMIBC [36]. Intravesical chemotherapy was administered to 34% of patients, prior to entering AS. The inclusion and exclusion criteria remained consistent with the pervious study. Among all periods, active treatment was performed in 80.6% cases with a 13.6% rate of stage progression and a 20.7% rate of grade progression. Notably, four patients progressed to MIBC. Although it was a retrospective study, the authors identified multiple factors that were associated with an increased risk of grade progression such as multiplicity, previous stage and grade, age and time since initial TURBT; however, none of these factors were found to be correlated with stage progression. 

Hurle et al. published five articles updating their series of patients with AS for NMIBC in the Bladder Cancer Italian Active Surveillance (BIAS) project [37,38,39]. They included patients with small (<10 mm), papillary, low-grade pTa or pT1 bladder cancer, with fewer than five lesions, negative cytology and no hematuria. Patients with a history of high-grade tumors or CIS were excluded. Follow-up included a cystoscopy and bladder-washing cytology every 3 months during the first year and every 6 months thereafter. The first publication included 55 patients with a median follow-up of 53 months [37]. Intravesical treatment (either Mitomycin C or BCG) was administered to 53 patients. Among the cohort, 51% discontinued AS and underwent TURBT, with a median AS duration of 12.5 months. The primary reasons for discontinuing AS were an increase in tumor size and/or number (53.6) followed by gross hematuria (32.1%) and. After discontinuation of AS, five patients had an upstage on the subsequent TURBT and three had an upgrade. Thus, conclusions of the authors were consistent with previous publications, concluding that AS can be a reasonable strategy in patients with small low-grade recurrent pTa and pT1 papillary tumors. 

A year later, Hurle et al. [38] updated their cohort to include 122 patients (146 AS events), with a median follow-up time of 11.9 months and a median AS duration of 11 months. The most common cause of discontinuation of AS were positive cytology (40.4%), followed by an increase in tumor number (27.1%) or size (27.1%), and gross hematuria (18.6%). No patients developed MIBC and the rates of grade or stage progression were negligible.

In their third publication, Hurle et al. [39] expanded the cohort to include 167 patients with 181 AS events with the same inclusion and exclusion criteria. The median follow-up time was 13 months. Of the 61 who discontinued AS, 20 had no malignancy on the subsequent histopathology specimen, 6 patients showed a high-grade disease and 5 had CIS. No cases of pT2 upstaging were observed. The authors identified no predicting factors of AS failure. They concluded that AS remains a viable strategy for patients with small, low-grade, recurrent pTa or pT1 bladder cancer. 

In 2021, the same research group published an update to the BIAS project database including 214 patients with 251 AS events [5], with a median follow-up duration of 38.8 months. Intravesical therapy, using either Mitomycin C or BCG, was administered to 35.8% of the patients. The median time on AS was 13 months, during which 51.8% experienced AS failure and underwent TURBT due to an increase in tumor size (39.2%), number of tumors (26.1%), positive cytology results (8.5%) or gross hematuria (2.3%). Among those patients who underwent TURBT, 25 cases (19.2%) revealed benign lesions on pathological examination, 12 cases exhibited high-grade tumors and one patient (0.7%) was diagnosed with T2 stage disease. The authors concluded that a history of several prior TURBTs and the presence of multiple lesions at the time of initial enrollment in AS were significantly associated with an increased risk of AS failure. 

Lokeshwar et al. conducted a retrospective analysis of 45 patients undergoing AS or office cautery for low-grade Ta recurrences [40]. Tumors were characterized as low-grade Ta, with a maximum size of <20 mm, and fewer than five lesions. The median follow-up period was 62 months. During this time, stage progression occurred in four patients (9%), while two patients required systemic therapy due to grade progression, one of whom subsequently developed MIBC. The authors concluded that AS and/or office cautery for patients with small, recurrent, low-grade pTa tumors is a safe strategy that can reduce the need for TURBTs. 

The most recent study on AS, published in 2023 by Tan et al. [41], prospectively analyzed 163 patients undergoing 208 AS periods. The median follow-up duration was 33 months. The study population comprised patients with fewer than five small (<10 mm) lesions and previous low-grade (G1 or G2) Ta or T1 tumors. Notably, only six patients (3.7%) had a history of T1 tumors, while the majority (97.3%) were classified as Ta. Over the course of AS, 66.9% of cases required TURBT due to an increase in tumor number and/or size (81.7%), positive cytology (6.3%), or the presence of gross hematuria (5.8%). Furthermore, progression in tumor grade was observed in 10 patients (6.1%), while stage progression occurred in 4 individuals (2.5%); however, no cases progressed to a MIBC. Follow-up involved bladder-washing cytology and cystoscopy every 3 months for the first year and then reduced to every 6 months thereafter. 

## 4. Discussion

In this systematic review, we identified 11 articles that met the PICO selection criteria for evaluating active surveillance in NMBIC. The available literature on AS is limited and predominantly published by a few research groups. Most studies were not randomized, observational and prospective, with the exception of those by Pruthi et al. [33], Finocchiaro et al. [43], and Lokeswar et al. [40] which were retrospective. The majority of studies applied similar inclusion criteria, focusing on patients with recurrent low-grade (G1 or G2) pTa or pT1 tumors. However, exceptions were noted in the two earliest publications, which also included G3 and CIS tumors [23,33]. Gofrit et al. excluded pT1 tumors, analyzing only patients with a history of low-grade pTa [34,43]. Additionally, Contieri et al. [25] and Hurle et al., in their studies [37,38,39], included patients with low-grade (G1 and G2) pTa and only pT1a tumors (involving the lamina propria but above the muscularis mucosa [44]). 

The median age of the patients included in the studies was approximately 70 years, with ages ranging from 39 to 88 years. This broad age range demonstrates that AS is not solely a treatment strategy reserved for elderly patients but should carefully considered for selected younger patients, it is important to recognize that the risk of progression, although low, is not negligible. Therefore, a personalized approach is crucial, balancing the advantages of avoiding surgical intervention against the long-term consequences associated with disease progression. In contrast, elderly patients or those with significant comorbidities may derive greater benefit from AS, as the risks related to anesthesia and surgery during TURBT are proportionally higher in this population.

Most studies enrolled patients with small tumor sizes (<10 mm), a limited number of tumors (<5), and negative urinary cytology. However, some variations in the inclusion criteria were noted. Lokeshwar et al. included patients with lesions up to 20 mm in size [40] while Finocchiaro et al. extended the size criterion up to 30 mm [43]. Soloway et al. [23] did not specify a tumor size but included patients with small tumors and up to four lesions. Similarly, Pruthi et al. did not specify any criteria regarding tumor size or the number of tumors [33]. Notably, none of the studies employed a standardized technique for precise tumor measurement. 

In most studies, approximately half of the cohort were subjected to some type of intravesical therapy (either BCG or Mitomycin C) after the first TURBT. However, in the study by Finocchiaro et al. only 28% of the population received intravesical therapy [43], whereas Hurle et al. reported that 96% of their patients received intravesical treatment before starting AS [37]. 

The median follow-up of the study ranged from 11.9 months to 73 months, while the median duration of AS varied from 9 to 13.4 months. The most common reason for discontinuing AS was an increase in tumor size and/or number, followed by positive cytology and the presence of hematuria. The rate of AS failure varied widely among studies, ranging from 36.5% in the cohort reported by Hurle et al. [38], to 80.6% in the study by Hernández et al. [36]. All studies discontinued AS and performed TRUBT or office fulguration when there was an increase in tumor size, tumor number, hematuria, a positive cytology or a voluntary withdrawal from the study.

One of the main limitations of AS is the need for frequent cystoscopies, which can be uncomfortable, costly, and burdensome for patients. Although cystoscopy remains the gold-standard for bladder tumor detection, there has been interest in exploring less invasive follow-up strategies. In certain protocols, ultrasound imaging and urinary biomarkers have been used intermittently incorporated alongside cystoscopy, particularly for patients with low-risk disease or when cystoscopy is poorly tolerated [45,46]. However, ultrasound has lower sensitivity for detecting flat or very small bladder tumors and cannot reliably replace cystoscopy in routine surveillance. As of now, there is insufficient evidence to support ultrasound as a sole follow-up tool in AS protocols, but its use as an adjunct may offer some benefit in selected cases or settings where cystoscopy is not feasible. 

To address the invasiveness of cystoscopies, several urinary biomarkers are under investigation for their potential to safely reduce the frequency of invasive procedures. Biomarkers such as UroVysion and Xpert Bladder Cancer Monitor, both of which demonstrate high negative predictive values, have shown promise in identifying patients at low-risk of recurrence [47]. Their integration into AS protocols could enable a more personalized approach to follow-up, potentially allowing for the extension of cystoscopy intervals in carefully selected cases. Nonetheless, prospective validation in AS-specific cohorts remains essential before these biomarkers can be routinely implemented to replace or defer cystoscopies. 

The follow-up protocol for patients under AS has been a topic of debate for the last years. However, the majority of the included studies applied a similar follow-up approach, conducting a bladder-washing cytology and cystoscopies every 3 months during the first year, followed by every 6 months thereafter. Nonetheless, Gofrit et al. performed cystoscopies every 3 months during the first 2 years and subsequently every 6 months [34]. Moreover, Hernández et al. in their initial study, performed a bladder-washing urine cytology and a flexible cystoscopy every 3–4 months until AS was discontinued [35]. Only three studies did not specify the use of urinary cytology in their follow-up protocols with alongside cystoscopy; nevertheless, all of them required a negative cytology as part of their inclusion criteria. 

Imaging of the upper urinary tract was performed according to the European Association of Urology guideline recommendations [48]. 

Another factor that may influence future indications for AS is the growing adoption of photodynamic diagnosis-assisted TURBT (PDD-TURBT) [49]. Several studies have shown that PPD-TURBT improves tumor detection and significantly reduces recurrence rates, particularly in patients with NMIBC [50]. By enhancing the completeness of tumor resection, PDD may lower the recurrence burden, which is one of the key drivers for considering AS. Nevertheless, AS may still remain a reasonable option for selected patient populations, particularly those with multiples comorbidities, limited life expectancy, or high surgical risk. 

The primary endpoints of all studies were to evaluate the oncological outcomes of AS and to assess the risks of stage and grade progression. Grade progression rates was ranged from 20.7% in the Hernandez et al. cohort [36] to no patients in the Gofrit et al. study [34]. This difference may be attributed to the exclusion of patients with pT1 tumors in the Gofrit et al. cohort [34], unlike in the other studies. Similarly, stage progression was most frequent in the study by Hernández et al. [36], with a rate of 13.6% while Gofrit et al. documented no cases [34]. Out of the 1.354 observational periods and 1.101 patients included in the analysis, only six cases of MIBC were identified and reported only in three studies [25,36,40]. Hernández et al. recorded the highest incidence, with four individuals (2.15%) developing MIBC [36] (all initially diagnosed with pT1G2 tumors <3 mm when they entered AS). In contrast, a single case of muscle-invasive progression was noted in the cohorts by Contieri et al. [5], Finocchiaro et al. [43], and Lokeshwar et al. [40]. Even though Soloway [23] and Pruthi [33] included patients with high-grade disease and CIS, no progression to MIBC was reported. This absence of progression may be attributable to the small sample sizes in their studies, with only 32 and 22 patients, respectively. 

This study has some notable limitations. On the first hand, there is a substantial heterogeneity among the included studies, particularly regarding the inclusion and exclusion criteria, follow-up protocols and surveillance failure definitions. These variations complicate direct comparisons and limit the generalizability of the results, thus in order to overcome them there is a pressing need for international collaborative efforts to develop harmonized guidelines for AS in NMIBC (Table 3). Consensus-based frameworks could help standardize patient selection, follow-up intervals, and criteria for discontinuation, facilitating more consistent clinical practice and enabling more robust comparative research. Ongoing prospective studies and clinical registries may play a key role in informing such efforts, ultimately helping to optimize patient outcomes while minimizing unnecessary interventions. Moreover, most of the included studies were single-arm, observational cohorts lacking control groups, which limits the strength of the conclusions regarding the comparative safety or efficacy of AS. The absence of randomized controlled trials or standardized outcome definitions underscores the need importance of interpreting current evidence with caution when applying it to routine clinical practice. 

On the other hand, there is a lack of comparative studies analyzing the quality of life between patients on AS and those who underwent TURBT. While both management strategies are used based on tumor characteristics and patient risk profiles, the psychological and physical impacts on patients may differ significantly. AS is intended to minimize overtreatment and procedural burden, but requires frequent cystoscopies that, besides being an invasive diagnostic tool may contribute to uncertainty, anxiety and stress regarding the risk of disease progression. Controversially, TURBT, while more invasive initially, may provide patients with a sense of therapeutic closure, reduce uncertainty and the number of cystoscopies. The absence of direct comparison between these strategies leaves a significant gap in understating how these management options influence patients’ overall well-being, which is essential for making fully informed treatment decisions. Further research is needed to address this subject in order to provide clearer insights into long-term quality of life outcomes for both patients cohorts.

Finally, another potential advantage of AS is its economic impact. By reducing the number of TURBT, hospitalizations, and anesthesia-related procedures, AS could significantly decrease healthcare costs, particularly in health systems under resource constraints. Evidence from AS in prostate cancer and small renal masses suggests substantial cost savings without compromising oncological outcomes. Although direct cost-effectiveness analyses comparing AS with standard TURBT management in NMIBC are currently lacking, preliminary data indicate that similar economic benefits may be achievable. Future prospective studies are needed to systematically evaluate the economic implications of AS in this specific patient population. In addition, the development of reliable urinary biomarkers, such as miRNA profiles, could enhance the safety and efficiency of AS by improving patient selection and reducing reliance on frequent invasive cystoscopies.

## 5. Conclusions

This systematic review confirms that active surveillance is a feasible and safe management strategy for a well-defined subset of patients with low-grade, non-muscle-invasive bladder cancer. While the current literature remains limited by heterogeneity and a lack of randomized controlled trials, our analysis highlights that AS is associated with low rates of progression to high-grade or muscle-invasive disease when applied to carefully selected patients. Importantly, the findings support that AS may reduce the surgical burden without compromising oncologic safety in specific populations, particularly the elderly and those with comorbidities. These insights reinforce the need to develop standardized inclusion criteria and follow-up protocols, and to design future prospective studies focused not only on oncological outcomes but also on quality of life and healthcare resource utilization. 

## Figures and Tables

**Figure 1 cancers-17-01714-f001:**
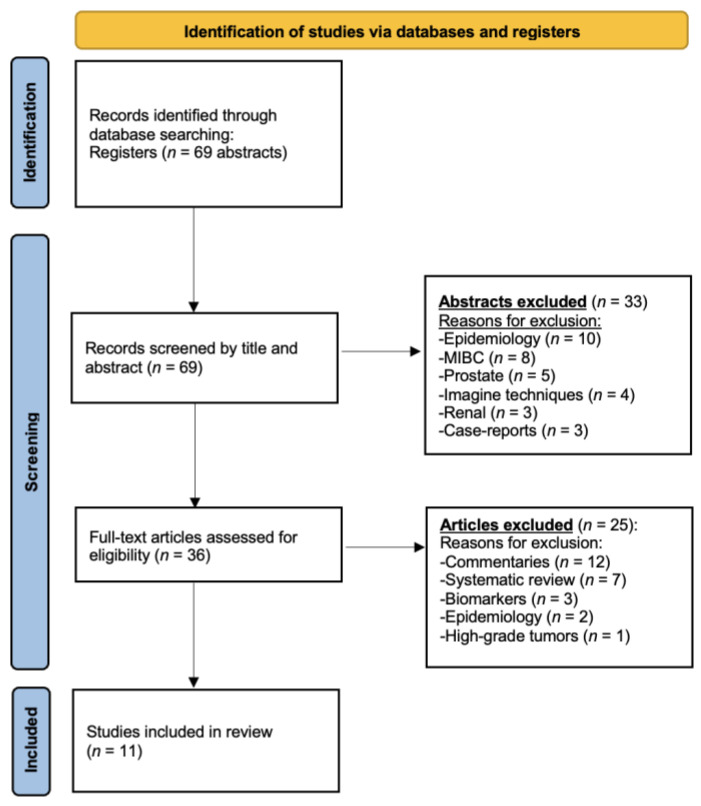
PRISMA flowchart.

**Table 1 cancers-17-01714-t001:** Newcastle–Ottawa quality assessment scale to analyze the quality of the included studies.

	Selection				Comparability	Outcome		
**Study**	Representativeness of exposed cohort	Selection of non-exposed cohort	Ascertai-nment of exposure	Demonstration that outcome of interest was not present at start	Design or analysis	Ascertainment of outcome	Adequacy of follow-up	Adequacy of follow-up of cohorts
Soloway, 2003 [23]	(+)	(+)	(+)	(+)	(+)	(+)	(+)	(-)
Pruthi, 2008 [33]	(+)	(-)	(+)	(+)	(-)	(+)	(+)	(-)
Gofrit, 2008 [34]	(+)	(-)	(+)	(+)	(-)	(+)	(+)	(+)
Hernández, 2009 [35]	(+)	(+)	(+)	(+)	(+)	(+)	(+)	(+)
Hernández, 2016 [36]	(+)	(-)	(+)	(+)	(-)	(+)	(+)	(-)
Hurle, 2016 [37]	(+)	(-)	(+)	(+)	(-)	(+)	(+)	(-)
Hurle, 2017 [38]	(+)	(-)	(+)	(+)	(-)	(+)	(+)	(-)
Hurle, 2018 [39]	(+)	(-)	(+)	(+)	(-)	(+)	(+)	(-)
Contieri, 2022 [25]	(+)	(-)	(+)	(+)	(+)	(+)	(+)	(+)
Lokeshwar, 2022 [40]	(+)	(-)	(+)	(+)	(+)	(+)	(+)	(+)
Tan, 2023 [41]	(+)	(-)	(+)	(+)	(+)	(+)	(+)	(+)

Risk of bias: (-) High.
(+) Low.

**Table 2 cancers-17-01714-t002:** Overview of studies included in this systematic review evaluating AS for NMBIC.

Author, Year	Patients/ AS Periods	Type of Study	Inclusion Criteria	Pathological Finding Before AS	Previous Intravesical Therapy (%)	Median Follow-Up (Months)	Median AS (Months)	AS Failure Rate (%)	Grade Progression *n* (%)	Stage Progression *n* (%)	Progression to MIBC (%)	Follow-Up	Discontinuation AS	Exclusion Criteria
Soloway MS, 2003 [23]	32/56	Retrospective	Small (no size reported), <4 lesions	Ta/G1/G2/G3/T1/CIS	53	38	10.1	100	9.4	6.3	0	Cystoscopy every 3–5 months		
Pruthi R, 2008 [33]	22/32	Retrospective	LG or HG papillary urothelial carcinoma or CIS	Ta/T1/CIS/G1/G2/G3	68.8	25		31.8	4.5	9	0	Cystoscopy every 3 months for 2 years and every 6 months until 5 years and then annually		Urothelial papilloma, PULMP
Gofrit O, 2008 [34]	31/43	Retrospective	10 mm, negative UC	Ta/G1/G2	58.1	16.1		81.4	3.1%	3.1	0		Increase in number and/or size (62.8%), patient’s request (16.3%)	
Hernández V, 2009 [35]	64/70	Prospective	<10 mm <5 tumors and negative UC	Ta/T1/G1/G2	NA	38.6	10.3	75.8	16.2	6.5	0	UC + cystoscopy every 4 months for 1 year, then alternating US and cystoscopy + UC every 6 months	Increase in number and/or size (58.6%), hematuria (4.3%), positive UC (2.9%)	History of HG carcinoma (G3), CIS or positive UC
Hernández V, 2016 [36]	186/252	Prospective		Ta/T1/G1/G2	43	72	13.4	80.6	20.7	13.6	2.15	UC + cystoscopy every 4 months for 2 years, followed by every 6 months (alternating cystoscopy and US)	Increase in number and/or size (61.9%), increase in number and/or size and positive UC (8.7%), positive UC (7.1%)	
Hurle R, 2016 [37] ^BIAS project^	55/70	Prospective		Ta/**T1a**/G1/G2	96.4	53	12.5	51	18	13.2	0	UC + cystoscopy every 3 months for 1 year, then every 6 months	Increase in number and/or size (53.8%), hematuria (32%), positive UC (14.3%)	
Hurle R, 2017 [38] ^BIAS project^	122/146	Prospective		Ta/**T1a**/G1/G2	41.0	11.9	11	37.7	13.1	7.4	0		Increase in number and/or size (64.4%), hematuria (18.6%), positive UC (16.9%)	History of HG carcinoma (G3), CIS or positive UC
Hurle R, 2018 [39] ^BIAS project^	167/181	Prospective		Ta/**T1a**/G1/G2	36.5%	13	12	36.5	NA	NA	0		Increase in number and/or size (22.1%), hematuria (6.1%), positive UC (5.5%)	
Contieri R, 2022 [25] ^BIAS project^	214/251	Prospective		Ta/**T1a**/G1/G2	35.8%	38.8	13	51.8	NA	NA	0.7		Increase in number and/or size (87.7%), hematuria (2.3%), positive UC (8.5%)	
Lokeshwar SD, 2022 [40]	45	Retrospective	<5 lesions, <2 cm in prior TURBT, LGTa appearance	Ta/T1/G1/G2	58	62	NA	89	11	9	2.2	Cystoscopy every 3–6 months		CIS or MIBC
Tan WS, 2023 [41] ^BIAS project^	163/208	Prospective	≤5 lesions, <1 cm in prior TURBT, LGTa appearance	Ta/T1/G1/G2	NA	33		66.9	6.1	2.5	0	UC + cystoscopy every 3 months for 1 year, then every 6 months	Increase in number and/or size (81.7%), positive UC (6.3%) and hematuria (5.8%)	

AS: active surveillance; CIS: carcinoma in situ; HG: high-grade; LG: low-grade; MIBC: muscle-invasive bladder cancer; NA: not available; PULMP: papillary urothelial of low malignant potential; TURBT: transurethral resection of bladder tumor; UC: urinary cytology; US: ultrasound.

**Table 3 cancers-17-01714-t003:** Proposed clinical criteria for AS in NMIBC.

Criterion	Characteristics for AS
Tumor size	≤3 cm
Number of tumors	≤5 tumors
Tumor appearance	Papillary
Urinary cytology	Negative
Previous pathology	No history of high-grade disease (G3) or carcinoma in situ (CIS)
Symptoms	Absence of gross hematuria
Follow-up compliance	Willing and able to adhere to cystoscopy follow-up

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
