# Peer review of "Active Surveillance in Non-Muscle Invasive Bladder Cancer: A Systematic Review"

_cancers, 2025, doi:10.3390/cancers17101714_

Round 1

Reviewer 1 Report

Comments and Suggestions for Authors

This manuscript reviews previously published articles regarding the clinical efficacy of active surveillance of noninvasive bladder urothelial cancer. The authors did not provide novel conclusion on this analysis.

Author Response

For review article

Response to Reviewer 1 Comments

1. Summary

2. Questions for General Evaluation

Reviewer’s Evaluation

Are all the cited references relevant to the research?

Yes/Can be improved/Must be improved/Not applicable

Is the research design appropriate?

Yes/Can be improved/Must be improved/Not applicable

Are the methods adequately described?

Yes/Can be improved/Must be improved/Not applicable

Are the results clearly presented?

Yes/Can be improved/Must be improved/Not applicable

Are the conclusions supported by the results?

Yes/Can be improved/Must be improved/Not applicable

3. Point-by-point response to Comments and Suggestions for Authors

Comments 1:

This manuscript reviews previously published articles regarding the clinical efficacy of active surveillance of noninvasive bladder urothelial cancer. The authors did not provide novel conclusion on this analysis.

Response 1:

Thank you for pointing this out. We completely agree with this comment. Therefore, we changed the entire conclusion:

“This systematic review confirms that active surveillance is a feasible and safe management strategy for a well-defined subset of patients with low-grade, non-muscle-invasive bladder cancer. While the current literature remains limited by heterogeneity and a lack of randomized controlled trials, our analysis highlights that AS is associated with low rates of progression to high-grade or muscle-invasive disease when applied to carefully selected patients. Importantly, the findings support that AS may reduce the surgical burden without compromising oncologic safety in specific populations, particularly the elderly and those with comorbidities. These insights reinforce the need to develop standardized inclusion criteria and follow-up protocols, and to design future prospective studies focused not only on oncological outcomes but also on quality of life and healthcare resource utilization.”

4. Response to Comments on the Quality of English Language

Point 1:

(x) The English is fine and does not require any improvement.

5. Additional clarifications

Reviewer 2 Report

Comments and Suggestions for Authors

This systematic review tackles a subject of increasing relevance in uro-oncology – the potential role of active surveillance (AS) in managing recurrent, low-grade non–muscle invasive bladder cancer (NMIBC). The concept is attractive, and this manuscript arrives at a moment when many urologists are re-evaluating intervention thresholds, especially in older or comorbid patients where surgical burden matters greatly.

Overall, the review is well constructed, methodologically sound, and offers a fairly comprehensive summary of the literature. That said, a few issues – both structural and interpretative – warrant attention prior to publication.

The authors are to be commended for gathering data across a spectrum of studies, many of which originate from centres that have led the AS conversation in NMIBC. However, the manuscript could benefit from a firmer critical tone, particularly in light of the methodological heterogeneity and lack of high-level evidence underpinning current practice.

It would be helpful to expand the introduction slightly to better frame why AS matters beyond economic or procedural burden. Recent work has begun to reframe low-grade NMIBC as a biologically distinct entity, rather than merely a stage-grade definition. Sex-related biological differences, as well as emerging molecular markers (e.g., urinary miRNA clusters), are contributing to a more nuanced stratification of risk in low-grade disease. The authors may wish to subtly acknowledge this evolving perspective, even if molecular aspects are not covered in this review per se.

The authors correctly point out that quality-of-life (QoL) data are lacking. However, this limitation is perhaps underplayed. It could be worth underscoring that, paradoxically, while AS aims to be less invasive, repeated cystoscopies may actually contribute to anxiety and psychological burden – an issue that has been raised in prostate and kidney AS literature and may well apply here. TURBT, although more aggressive, at least offers temporary procedural closure. This trade-off deserves more reflection.

The authors provide a reasonable summary of the Newcastle-Ottawa scores. However, they might consider stating more explicitly that most of the included data is from single-arm observational cohorts – and that therefore conclusions about comparative safety or efficacy must be interpreted with considerable caution. A sentence to this effect in the Discussion would improve balance.

There is mention of heterogeneity, but no real proposal for how this could be overcome in future work. It might be worth suggesting that international efforts could aim at generating harmonised criteria for AS eligibility, follow-up intervals, and discontinuation triggers. A mention of ongoing efforts or a call to build such consensus would be timely.

The literature search is said to follow PRISMA guidelines, which is appropriate, but the PRISMA flowchart was not seen in the version provided. Ensure this is included in the final submission.

Consider including a schematic summary table or figure highlighting which inclusion/exclusion criteria were shared vs divergent across the included studies. It would help readers digest the variability more intuitively.

Author Response

Response to Reviewer 2 Comments

1. Summary

2. Questions for General Evaluation

Reviewer’s Evaluation

Are all the cited references relevant to the research?

Yes/Can be improved/Must be improved/Not applicable

Is the research design appropriate?

Yes/Can be improved/Must be improved/Not applicable

Are the methods adequately described?

Yes/Can be improved/Must be improved/Not applicable

Are the results clearly presented?

Yes/Can be improved/Must be improved/Not applicable

Are the conclusions supported by the results?

Yes/Can be improved/Must be improved/Not applicable

3. Point-by-point response to Comments and Suggestions for Authors

This systematic review tackles a subject of increasing relevance in uro-oncology – the potential role of active surveillance (AS) in managing recurrent, low-grade non–muscle invasive bladder cancer (NMIBC). The concept is attractive, and this manuscript arrives at a moment when many urologists are re-evaluating intervention thresholds, especially in older or comorbid patients where surgical burden matters greatly.

Overall, the review is well constructed, methodologically sound, and offers a fairly comprehensive summary of the literature. That said, a few issues – both structural and interpretative – warrant attention prior to publication.

Comments 1:

The authors are to be commended for gathering data across a spectrum of studies, many of which originate from centers that have led the AS conversation in NMIBC. However, the manuscript could benefit from a firmer critical tone, particularly in light of the methodological heterogeneity and lack of high-level evidence underpinning current practice.

Response 1:

Thank you for pointing this out. We completely agree with this comment. Therefore, we changed the second paragraph on page 13 into:

“This study has some notable limitations. On the first hand, there is a substantial heterogeneity among the included studies, particularly regarding the inclusion and exclusion criteria, follow-up protocols and surveillance failure definitions. These variations complicate direct comparisons and limit the generalizability of the results, thus in order to overcome them there is a pressing need for international collaborative efforts to develop harmonized guidelines for AS in NMIBC. Consensus-based frameworks could help standardize patient selection, follow-up intervals, and criteria for discontinuation, facilitating more consistent clinical practice and enabling more robust comparative research. Ongoing prospective studies and clinical registries may play a key role in informing such efforts, ultimately helping to optimize patient outcomes while minimizing unnecessary interventions. Moreover, most of the included studies were single-arm, observational cohorts lacking control groups, which limits the strength of the conclusions regarding the comparative safety or efficacy of AS. The absence of randomized controlled trials or standardized outcome definitions underscores the need importance of interpreting current evidence with caution when applying it to routine clinical practice.”

Comments 2:

It would be helpful to expand the introduction slightly to better frame why AS matters beyond economic or procedural burden. Recent work has begun to reframe low-grade NMIBC as a biologically distinct entity, rather than merely a stage-grade definition. Sex-related biological differences, as well as emerging molecular markers (e.g., urinary miRNA clusters), are contributing to a more nuanced stratification of risk in low-grade disease. The authors may wish to subtly acknowledge this evolving perspective, even if molecular aspects are not covered in this review per se.

Response 2:

We completely agree with this comment this is why we revised the article in order to emphasize this point and added the following paragraph on page 2 of the introduction:

“Recent advances in the molecular understanding of bladder cancer suggest that low-grade, non-muscle-invasive tumors represent a biologically distinct entity with a separate pathogenesis from high-grade tumors. The biological characteristics of the initial tumor remain the most significant predictor of long-term outcomes[1]. The low-grade pathway is thought to originate from benign urothelium through a process of urothelial hyperplasia. In contrast, the majority of invasive tumors are likely to originate from the progression of dysplasia to carcinoma in situ and papillary high-grade noninvasive or invasive cancer[2]. This distinction could explain why low-grade tumor recurrences have usually the same histological features and the difference in risk of progression between both groups[3]. Indeed, Catto et al. demonstrated that miRNA expression profiles share more alterations between high-grade NMIBC and MIBC than between low-grade and high-grade NMIBC tumors, suggesting that despite the absence of muscle-invasion at the time of TURBT, high-grade NMIBC is a clear precursor to muscle-invasive disease and thus warrants a more aggressive and intensive therapeutic approach akin to that used for MIBC[4].” 

Comments 3:

The authors correctly point out that quality-of-life (QoL) data are lacking. However, this limitation is perhaps underplayed. It could be worth underscoring that, paradoxically, while AS aims to be less invasive, repeated cystoscopies may actually contribute to anxiety and psychological burden – an issue that has been raised in prostate and kidney AS literature and may well apply here. TURBT, although more aggressive, at least offers temporary procedural closure. This trade-off deserves more reflection.

Response 3:

We totally agree with this comment. Therefore, we changed the last paragraph of the discussion (page 13): 

“On the other hand, there is a lack of comparative studies analyzing the quality of life between patients on AS and those who underwent TURBT. While both management strategies are used based on tumor characteristics and patient risk profiles, the psychological and physical impacts on patients may differ significantly. AS is intended to minimize overtreatment and procedural burden, but requires frequent cystoscopies that besides being an invasive diagnostic tool may contribute to uncertainty, anxiety and stress regarding the risk of disease progression. Controversially, TURBT, while more invasive initially, may provide patients with a sense of therapeutic closure, reduce uncertainty and the number of cystoscopies. The absence of direct comparison between these strategies leaves a significant gap in understating how these management options influence patients’ overall well-being, which is essential for making fully informed treatment decisions. Further research is needed to address this subject in order to provide clearer insights into long-term quality of life outcomes for both patients cohorts.”

Comments 4:

The authors provide a reasonable summary of the Newcastle-Ottawa scores. However, they might consider stating more explicitly that most of the included data is from single-arm observational cohorts – and that therefore conclusions about comparative safety or efficacy must be interpreted with considerable caution. A sentence to this effect in the Discussion would improve balance.

There is mention of heterogeneity, but no real proposal for how this could be overcome in future work. It might be worth suggesting that international efforts could aim at generating harmonized criteria for AS eligibility, follow-up intervals, and discontinuation triggers. A mention of ongoing efforts or a call to build such consensus would be timely.

Response 4:

Thank you for pointing this out. We completely agree with your comment. We changed the second paragraph on page 13 in order to include both, the fact that the majority of studies were single-arm and a proposal for future work:

“This study has some notable limitations. On the first hand, there is a substantial heterogeneity among the included studies, particularly regarding the inclusion and exclusion criteria, follow-up protocols and surveillance failure definitions. These variations complicate direct comparisons and limit the generalizability of the results, thus in order to overcome them there is a pressing need for international collaborative efforts to develop harmonized guidelines for AS in NMIBC. Consensus-based frameworks could help standardize patient selection, follow-up intervals, and criteria for discontinuation, facilitating more consistent clinical practice and enabling more robust comparative research. Ongoing prospective studies and clinical registries may play a key role in informing such efforts, ultimately helping to optimize patient outcomes while minimizing unnecessary interventions. Moreover, most of the included studies were single-arm, observational cohorts lacking control groups, which limits the strength of the conclusions regarding the comparative safety or efficacy of AS. The absence of randomized controlled trials or standardized outcome definitions underscores the need importance of interpreting current evidence with caution when applying it to routine clinical practice.”

Comment 5:

The literature search is said to follow PRISMA guidelines, which is appropriate, but the PRISMA flowchart was not seen in the version provided. Ensure this is included in the final submission.

Response 5:

Thank you for pointing this out. The PRISMA flowchart is added into the review.  

4. Response to Comments on the Quality of English Language

Point 1:

(x) The English is fine and does not require any improvement.

5. Additional clarifications

Reviewer 3 Report

Comments and Suggestions for Authors

general comments

In this study, the authors review active surveillance of non-muscle invasive bladder cancer. This is interesting because active surveillance for bladder cancer is less common than for prostate and kidney cancer. Active surveillance may be a potential strategy for bladder cancer, but the benefits of active surveillance are unclear and it has several limitations. Please note that there are several issues that need to be addressed in this manuscript.

Specific comments for revision:

  1. Although the authors emphasize the potential dangers of TUR-BT, the risk of TUR-BT is low for small bladder cancers for which active surveillance is indicated. Elderly patients, those with comorbidities, and those in poor general condition would likely benefit more from avoiding TUR-BT because of the high risks of anesthesia and surgery. Although low-risk bladder cancer has a low risk of progression, the risk of progression still exists and active surveillance should be avoided, especially in younger patients. Please consider the above.

  1. In recent years, it has been reported that photodynamic diagnosis-assisted TUR-BT can reduce the recurrence rate of bladder cancer. Although the authors state that there is a high risk of intravesical recurrence after TUR-BT, the widespread use of PDD-TURBT may reduce the risk of intravesical recurrence, which may affect the future indications for active surveillance. Please consider mentioning this procedure in the text.

  1. Most studies have performed frequent cystoscopies, and there is no benefit to be gained other than avoiding TUR-BT. Although it may be difficult depending on the location and shape of the tumor, if some cystoscopies could be substituted with non-invasive ultrasound examinations, the benefit to patients would be great. Are there any reports on alternative examinations to cystoscopies?

  1. On line 15, does NIMBIC mean NMIBC? Please correct it.

  1. There is no period at the end of line 216. Please correct it.

Author Response

Response to Reviewer 3 Comments

1. Summary

2. Questions for General Evaluation

Reviewer’s Evaluation

Are all the cited references relevant to the research?

Yes/Can be improved/Must be improved/Not applicable

Is the research design appropriate?

Yes/Can be improved/Must be improved/Not applicable

Are the methods adequately described?

Yes/Can be improved/Must be improved/Not applicable

Are the results clearly presented?

Yes/Can be improved/Must be improved/Not applicable

Are the conclusions supported by the results?

Yes/Can be improved/Must be improved/Not applicable

3. Point-by-point response to Comments and Suggestions for Authors

In this study, the authors review active surveillance of non-muscle invasive bladder cancer. This is interesting because active surveillance for bladder cancer is less common than for prostate and kidney cancer. Active surveillance may be a potential strategy for bladder cancer, but the benefits of active surveillance are unclear and it has several limitations. Please note that there are several issues that need to be addressed in this manuscript.

Comments 1:

Although the authors emphasize the potential dangers of TUR-BT, the risk of TUR-BT is low for small bladder cancers for which active surveillance is indicated. Elderly patients, those with comorbidities, and those in poor general condition would likely benefit more from avoiding TUR-BT because of the high risks of anesthesia and surgery. Although low-risk bladder cancer has a low risk of progression, the risk of progression still exists and active surveillance should be avoided, especially in younger patients. Please consider the above.

Response 1:

Thank you for pointing this out. We completely agree with this comment. Therefore, we changed the second paragraph on page 11 and added the following lines:

“The median age of the patients included in the studies was approximately 70 years, with ages ranging from 39 to 88 years. This broad age range demonstrates that AS is not solely a treatment strategy reserved for elderly patients but should carefully considered for selected younger patients, it is important to recognize that the risk of progression, although low, is not negligible. Therefore, a personalized approach is crucial, balancing the advantages of avoiding surgical intervention against the long-term consequences associated with disease progression. In contrast, elderly patients or those with significant comorbidities may derive greater benefit from AS, as the risks related to anesthesia and surgery during TURBT are proportionally higher in this population.”

Comments 2:

In recent years, it has been reported that photodynamic diagnosis-assisted TUR-BT can reduce the recurrence rate of bladder cancer. Although the authors state that there is a high risk of intravesical recurrence after TUR-BT, the widespread use of PDD-TURBT may reduce the risk of intravesical recurrence, which may affect the future indications for active surveillance. Please consider mentioning this procedure in the text.

Response 2:

Agree. We have, accordingly, revised the article in order to emphasize this point. This is why we added the following paragraph on page 12, paragraph 4:

“Another factor that may influence future indications for AS is the growing adoption of photodynamic diagnosis-assisted TURBT (PDD-TURBT)[5]. Several studies have shown that PPD-TURBT improves tumor detection and significantly reduces recurrence rates, particularly in patients with NMIBC[6]. By enhancing the completeness of tumor resection, PDD may lower the recurrence burden, which is one of the key drivers for considering AS. Nevertheless, AS may still remain a reasonable option for selected patient populations, particularly those with multiples comorbidities, limited life expectancy, or high surgical risk.”

Comments 3:

Most studies have performed frequent cystoscopies, and there is no benefit to be gained other than avoiding TUR-BT. Although it may be difficult depending on the location and shape of the tumor, if some cystoscopies could be substituted with non-invasive ultrasound examinations, the benefit to patients would be great. Are there any reports on alternative examinations to cystoscopies?

Response 3:

We agree with this comment. Therefore, we added a new paragraph on page 12 (the second one):

“One of the main limitations of AS is the need for frequent cystoscopies, which can be uncomfortable, costly, and burdensome for patients. Although cystoscopy remains the gold-standard for bladder tumor detection, there has been interest in exploring less invasive follow-up strategies. In certain protocols, ultrasound imaging and urinary biomarkers have been used intermittently incorporated alongside cystoscopy, particularly for patients with low-risk disease or when cystoscopy is poorly tolerated[7,8]. However, ultrasound has lower sensitivity for detecting flat or very small bladder tumors and cannot reliably replace cystoscopy in routine surveillance. As of now, there is insufficient evidence to support ultrasound as a sole follow-up tool in AS protocols, but its use as an adjunct may offer some benefit in selected cases or settings where cystoscopy is not feasible.

To address the invasiveness of cystoscopies, several urinary biomarkers are under investigation for their potential to safely reduce the frequency of invasive procedures. Biomarkers such as UroVysion and Xpert Bladder Cancer Monitor, both of which demonstrate high negative predictive values, have shown promise in identifying patients at low-risk of recurrence[9]. Their integration into AS protocols could enable a more personalized approach to follow-up, potentially allowing for the extension of cystoscopy intervals in carefully selected cases. Nonetheless, prospective validation in AS-specific cohorts remains essential before these biomarkers can be routinely implemented to replace or defer cystoscopies.”

Comments 4:

On line 15, does NIMBIC mean NMIBC? Please correct it.

Response 4:

Thank you for pointing this out. The error was corrected.

Comment 5:

There is no period at the end of line 216. Please correct it.

Response 5:

Thank you for pointing this out. The error was corrected.

4. Response to Comments on the Quality of English Language

Point 1:

(x) The English is fine and does not require any improvement.

5. Additional clarifications

Reviewer 4 Report

Comments and Suggestions for Authors

The research was well conducted and well presented.

Before considering for publication, in order to improve the quality of the paper, in my opinion:

  • it could be useful to add economic analysis and comparison of different approaches (AS vs. standard), and how AS could be improved by biomarkers ;
  • after the evidence analysis, it could be useful and of interest to draw up a sort of summary of all indications to AS

Author Response

For review article

Response to Reviewer 4 Comments

1. Summary

2. Questions for General Evaluation

Reviewer’s Evaluation

Are all the cited references relevant to the research?

Yes/Can be improved/Must be improved/Not applicable

Is the research design appropriate?

Yes/Can be improved/Must be improved/Not applicable

Are the methods adequately described?

Yes/Can be improved/Must be improved/Not applicable

Are the results clearly presented?

Yes/Can be improved/Must be improved/Not applicable

Are the conclusions supported by the results?

Yes/Can be improved/Must be improved/Not applicable

3. Point-by-point response to Comments and Suggestions for Authors

The research was well conducted and well presented.

Comments 1:

It could be useful to add economic analysis and comparison of different approaches (AS vs. standard), and how AS could be improved by biomarkers

Response 1:

Thank you for pointing this out. We completely agree with this comment. Therefore, we changed the second paragraph on page 13 and added the following paragraph at the end of the discussion:

“Finally, another potential advantage of AS is its economic impact. By reducing the number of TURBT, hospitalizations, and anesthesia-related procedures, AS could significantly decrease healthcare costs, particularly in health systems under resource constraints. Evidence from AS in prostate cancer and small renal masses suggests substantial cost savings without compromising oncological outcomes. Although direct cost-effectiveness analyses comparing AS with standard TURBT management in NMIBC are currently lacking, preliminary data indicate that similar economic benefits may be achievable. Future prospective studies are needed to systematically evaluate the economic implications of AS in this specific patient population. In addition, the development of reliable urinary biomarkers, such as miRNA profiles, could enhance the safety and efficiency of AS by improving patient selection and reducing reliance on frequent invasive cystoscopies.”

Comments 2:

After the evidence analysis, it could be useful and of interest to draw up a sort of summary of all indications to AS.

Response 2:

Agree. We have, accordingly, revised the article in order to emphasize this point. This is why we added the following summary of AS indications:  

Criterion

Characteristics for AS

Tumor size

≤3 cm

Number of tumors

≤5 tumors

Tumor appearance

Papillary

Urinary cytology

Negative

Previous pathology

No history of high-grade disease (G3) or carcinoma in situ (CIS)

Symptoms

Absence of gross hematuria

Follow-up compliance

Willing and able to adhere to cystoscopy follow-up

4. Response to Comments on the Quality of English Language

Point 1:

(x) The English is fine and does not require any improvement.

5. Additional clarifications

Round 2

Reviewer 1 Report

Comments and Suggestions for Authors

This manuscript did a meta-analysis of the clinical value of active surveillance on the non-muscle invasive bladder cancer (NMIBC). The authors reported that early studies, such demonstrated AS as a feasible alternative to TURBT, with low progression rates. Subsequent research confirmed its safety in selected patients, with tumor growth and positive cytology being the main reasons for intervention. More recent investigations, further supported AS as a viable strategy, highlighting the low risk of stage and grade progression and its potential to reduce surgical interventions.

The conclusion of this study does not provide new concept on the current knowledge of clinical management of NMIBC.